# Risk Stratification and Clinical Characteristics of Patients with Late Recurrence of Melanoma (>10 Years)

**DOI:** 10.3390/jcm11072026

**Published:** 2022-04-05

**Authors:** Robin Reschke, Konstantin Dumann, Mirjana Ziemer

**Affiliations:** Department of Dermatology, Allergology and Venereology, University Medical Center, 04103 Leipzig, Germany; robin.reschke@medizin.uni-leipzig.de (R.R.); konstantin.dumann@medizin.uni-leipzig.de (K.D.)

**Keywords:** melanoma, metastasis, prognosis

## Abstract

Background: Most patients with high-risk melanomas develop metastasis within the first few years after diagnosis. However, late recurrence of melanoma is seen in patients that metastasize more than 10 years after the primary diagnosis; a metastasis after 15 years is considered an ultra-late recurrence. It is critical to better understand the clinical and biological characteristics of this subset of melanoma patients in order to offer an individual treatment plan and prevent metastasis. Methods: We retrospectively analyzed melanoma patients with recurrence ≥10 years after a primary diagnosis documented between 1993 and 2012 at the skin cancer center of the University Medical Center Leipzig, Germany. We conducted a comprehensive review of the literature and compared the results with our data. Available archived primary melanoma tissue was investigated with a seven-marker immunohistochemical signature (immunoprint^®^) previously validated to reliably identify high-risk patients within stages IB-III. Results: Out of 36 analyzed patients, a third metastasized ultra-late (≥15 years). The mean age at initial diagnosis was 51 years, with women being diagnosed comparatively younger than men. The largest proportion of patients with late recurrence had primary melanomas located on the trunk. The immunoprint^®^ signature of the available primary melanomas allowed the accurate prediction of a high risk. However, it is difficult to draw a definitive conclusion from the small number of cases that could be analyzed with immunoprint^®^ (*n* = 2) in this study. Apart from the primary tumor characteristics, the laboratory values at time of metastasis, comorbidities and outcome are also shown. Conclusion: Late and ultra-late recurrent melanomas seem not to differ from melanomas in general, apart from a distinctly higher proportion of lower leg localizations in ultra-late recurrent melanomas. The immunoprint^®^ signature may help to identify high-risk primary tumors at the time of initial diagnosis. However, apart from the risk profile of the primary tumor, it seems that individual immune surveillance can control residual tumor cells for more than a decade. Advanced age and increasing comorbidities may contribute to a disturbed immunological balance.

## 1. Introduction

Melanoma incidence is continuously rising, and patients still die from metastatic disease, despite a portfolio of promising systemic treatment options [1,2]. Research still focuses on understanding the complex process of metastasis. A majority of patients with high-risk melanomas develop metastatic disease within the first three years after the diagnosis (65–85%), mostly in the first two years (55–67%) [3]. Breslow’s depth and the ulceration of the primary melanoma are the established histomorphologic parameters used for prognosis predictions. However, there is still a strong need for biomarkers that facilitate, at the time of primary diagnosis, a more precise identification of patients with a higher risk for recurrence [3,4,5]. Late recurrence of melanoma with metastasis ten years after the initial diagnosis illustrates that decades may pass between the detachment of individual tumor cells from the primary melanoma, their migration through blood and lymphatic vessels and the clinical manifestation of distant metastases. Unquestionably, in addition to histomorphologic characteristics, there seem to exist further factors influencing disease spread. It is a complex interplay of various factors that enables so-called “tumor dormancy”. The patient’s immune system plays an important role in regulating tumor growth with the secretion of interferon-γ and interleukin−2, among other substances [6,7]. Particular cellular mechanisms can also contribute to this phenomenon. Tumor cells can exclude themselves from the cell cycle and survive in a dormant state with distinctly reduced metabolic activity [6,7]. Another regulating mechanism is the angiogenic activity of tumor cells. In a stable state of cell division and death, the activation and inhibition of angiogenic factors are balanced. When this stable state shifts towards angiogenesis, tumor cells will grow (angiogenic switch) [7]. It is not only the cell’s intrinsic or immune-mediated mechanisms that influence the tumor cell’s survival but also environmental factors. Oxidative stress is a limiting factor for metastatic spread because it leads to the cell death of detached tumor cells. These rely on antioxidants such as glutathione or nicotinamide adenine dinucleotide phosphate (NADPH). Cells die without sufficient levels of these molecules [6,8]. Li et al. have shown that, among lung cancer patients, smokers show a better response to programmed cell death protein (PD)1/PD1-ligand 1-inhibitors than non-smokers [9]. One potential explanation could be the increased oxidative stress in patients who smoke [10]. A particular subpopulation of melanoma cells called cancer stem cells (CSC) or melanoma stem cells (MSC) is established during cellular stress [11]. These CD20+MSC can lead to tumor maintenance and migration. The group of melanoma patients with so-called late or ultra-late metastasis may help elucidate this concept of “tumor dormancy”.

## 2. Material and Methods

Included in the study were patients with late recurrent melanoma metastasizing ten or more years after the initial diagnosis of their primary tumors, as documented by the skin cancer center of the Department of Dermatology at the University Medical Center Leipzig, Germany, between 1993 and 2012. Initially, 38 patients fulfilled the inclusion criteria and were included in the study. Two patients developed a second primary melanoma before the onset of the metastatic disease and were excluded. Ultimately, 36 cases were analyzed. Twelve of the 36 patients (33.3%) metastasized more than 15 years after the primary melanoma, thus qualifying for so-called ultra-late recurrence. Collected clinical characteristics included the characteristics of the primary melanoma such as localization, the presence of ulceration and Breslow’s depth; the age at primary diagnosis and at the onset of metastatic disease; comorbidities; and outcome (Appendix A).

Laboratory results from S100ß protein and lactate dehydrogenase (LDH) as well as a differential blood count and thrombocytes were collected at the time of metastasis. Laboratory results from the time of primary diagnosis were not available. Clinical parameters were compared with 17 cohorts of late recurrent melanomas published in the literature between 1983 and 2021. Case reports were excluded. We additionally aimed to examine the available primary melanomas immunohistochemically with a 7-marker immunohistochemical signature, consisting of Bax, Bcl-X, PTEN, COX-2, (loss of) ß-Catenin, (loss of) MTAP and (presence of) CD20 (immunoprint^®^) [12]. Apart from three cases, however, the tumor blocks of the primaries more than 10 years after diagnosis did not exist. The main reason is the limit on archiving based on German archival storage guidelines. Of those cases, formalin-fixed and paraffin-embedded tissue sections were stained with the immunoprint^®^ signature. The determination and scoring of the signature were previously described in detail and validated in larger patient cohorts [12,13]. This analysis was conducted in cooperation with Synvie GmbH, Munich, Germany. The stained slides were evaluated by an independent external dermatologist and histopathologist (MD) and one experienced Synvie lab scientist (PhD) in a blinded manner.

## 3. Results

Our study cohort included equal gender representation: 18 male and 18 female patients. The mean age at the time of initial diagnosis was 51 years (30–76 years) (Table 1).

In more than half of the patients (*n* = 19, 52.8%), the primary melanoma was found on the trunk; in seven patients, it was on a lower extremity (19.4%); in five patients, it was on an upper extremity (13.9%); in another five patients, it was in the head and neck region (13.9%). Concerning the gender distribution, the primary melanoma in both male and female patients was predominantly localized at the trunk. However, in women, primaries on the lower and upper extremities, taken together, also comprised 44.4%. The period between the primary melanoma and metastasis was, on average, 15 years (10–33 years) (Table 1). The mean age of patients at the time of metastasis was 66 years. There was no significant difference between the organ manifestation of metastatic spread and the time intervals. Detailed information on the primary tumor was documented in 24 of 36 patients (66.7%). Half of these patients were male, and half were female. The average Breslow’s depth was 1.8 mm (0.4–4.5 mm, from pT1a to pT2b) (Table 2). In a third of patients (*n* = 12), documentation of detailed histopathological information for the primary melanoma was not available in the department’s database or that of the pre-treating physicians (*n* = 12) because of the long periods between primary melanoma and metastases.

Almost 17% of melanomas showed an ulceration (Table 2). There was no association between tumor thickness and time until metastasis. Tumor markers S100 and LDH were documented in two-thirds of the patients (66.7%) at the time of metastatic disease. The median S100 level was 0.07 ng/mL (0.05–2.6 ng/mL, mean 0.28 ng/mL, normal: <0.105 ng/mL) (Table 3).

In 42% of the male patients, the S100 value was increased, whereas the majority of women (60%) had an increased S100 value. The median value of LDH at the time of metastasis was 4.7 µkat/L (2.9–10.2 µkat, normal: <3.75 µkat/L) (Table 4). Increased LDH values at the time of metastatic disease were seen in the majority of patients independently of gender but more frequently in women (66.7% of men and 83.3% of women) (Table 4).

For 18 of 36 analyzed patients (50% women, 50% men) we had results of a differential blood count at the time of metastasis. The majority of patients had normal differential blood counts (Appendix A). Four patients showed leukocytosis and four patients lymphopenia (22.2%). Six patients presented with neutrophilia and one patient with eosinophilia. The neutrophil-to-lymphocyte ratio (NLR) was higher than 5 (normal: 0.78–3.53) in three women and three men (33.3% of all patients) [14]. Table 5 shows an association between a higher NLR and a longer period until the onset of metastatic disease.

In 20 of 36 patients (55.6%), information about comorbidities at the time of the initial diagnosis of the primary melanoma could be obtained. At the time of metastasis, this information was available from 33 of 36 patients (91.7%). The most frequently documented comorbidity was hypertension. At the time of initial diagnosis, hypertension was documented in eight patients (40%), compared to 23 patients at the time of metastatic disease (69.7%). Seven patients (21.2%) had a thromboembolic event (deep leg vein thrombosis, pulmonary artery embolism, myocardial infarction or ischemic apoplex) at the time of metastasis. At the time of primary diagnosis, only one patient had a documented ischemic apoplex. A non-cutaneous secondary tumor was diagnosed in four patients (16.7%) between the primary melanoma and the onset of metastatic disease (bladder cancer, lung cancer, intraductal papillary mucinous tumor of the pancreas and myeloproliferative neoplasm). At the time of the primary diagnosis, one patient had a renal cell carcinoma.

In Appendix A, the cases with ultra-late recurrence (metastases after more than 15 years) were compared to the cases with occurrences of metastases after 10–15 years. In total, 12 of 36 patients metastasized after more than 15 years (33.3%). Eight of these 12 patients were men. The age of patients at the time of initial diagnosis did not differ between patients with late and ultra-late recurrence of melanoma. Breslow’s depth and tumor markers (S100 and LDH) were also comparable between groups. The main location of the primary melanoma was different in late and ultra-late cases. In ultra-late recurrent patients, the majority of primary melanomas was on the lower extremities. Only a fourth of melanomas was located at the trunk. By contrast, in cases with late recurrence of melanoma, two-thirds of melanomas were found at the trunk and only five at the extremities.

Archival tissue from the primary tumor was limited to three cases. One of those was not evaluable due to immense melanin pigmentation (Appendix A). Of the two remaining, patient A was a female patient who had an ulcerated nodular melanoma with a Breslow’s depth of 2.6 mm located at the trunk. The period between the primary diagnosis and the onset of metastatic disease was eleven years. Subsequent staining with a previously validated immunohistochemical signature profile was able to stratify the patient with a high-risk score of 0.169 (<0.135 = low-risk; >0.135 = high-risk) and thereby correctly “predicted” the recurrence (Figure 1) [12,13]. Patient B was a female patient who had a superficial spreading melanoma with a Breslow’s depth of 0.9 mm located at the trunk. The period between the primary diagnosis and the onset of metastatic disease was ten years. Staining with the immunoprint^®^ profile was able to stratify the patient with a high-risk score of 0.200 and correctly “predicted” the recurrence as well (Figure 2).

## 4. Discussion

In summary, 36 patients with late recurrence of melanoma were analyzed. The mean age at diagnosis of the primary melanoma was 51 years. The mean time until the onset of metastatic disease was 15 years. At the time of the initial diagnosis, women were generally younger than men, which is consistent with the data of all melanoma cases documented in the German Cancer Registry [15]. Taking into account that the likelihood of late recurrence decreases in elderly patients, the age distribution of our cohort showed that patients older than 70 years were less represented. Comorbidities and natural death potentially anticipate possible late recurrences of melanoma. The gender distribution was equal among age groups. Men had more primary tumors at the trunk, whereas in women, they were mostly localized at the trunk but also on the extremities to nearly the same extent. This finding is consistent with the pattern of melanoma distribution described in the literature [16]. The mean Breslow’s depth of primary melanomas was 1.8 mm. In advanced disease stages, eosinophilia in peripheral blood is considered to be a good prognostic factor. It has been correlated with significantly better survival in melanoma patients [17]. On the other hand, a neutrophil-to-lymphocyte ratio (NLR) > 5 is considered a negative prognostic factor [18]. Patients in our study collective with a higher NLR showed a longer recurrence-free survival time (Table 5) but a worse outcome (9% vs. 17% overall survival). Additionally, LDH was increased in the majority of patients at metastasis. S100, however, was only increased in half of the patients (Table 3 and Table 4). This could argue for a higher sensitivity of LDH as opposed to S100 as a marker for disease progression. Nevertheless, in the literature, S100 is described as more sensitive towards newly occurring metastases [19]. Interestingly, in our patients, S100 and LDH at the time of metastatic disease were higher in women than in men. A limiting factor of this analysis is the relatively small samples size. A gender-specific predisposition has not been shown in the literature studies (Table 6).

The median age at the time of primary diagnosis was 51 years. The mean age of all late recurrence patients recorded in the literature and summarized in Table 6 was 45 years. The mean of the period between the primary diagnosis and the onset of metastatic disease is 15 years in our study; this is only half a year longer than in the corresponding literature. Mean Breslow’s depth is also comparable to the cited literature. Ultra-late recurrent melanomas were more frequently found in the lower extremity (41.7%), a more favorable localization in terms of prognosis [34,35]. Indeed, the EORTC nomogram—a valuable predictive score—includes anatomical location in addition to Breslow’s thickness and ulceration status [36]. The more favorable prognosis of limb melanomas might be explained by an ameliorated immune surveillance of melanoma cells in this location. Tsao et al. only focused on ultra-late recurrent melanomas (>15 years) in their study [27]. The lower mean age of their patients at primary diagnosis (36 years) is remarkable (Table 6). Ultra-late metastasizing patients in our own dataset had a mean age of 50 years at primary diagnosis, comparable to that of patients with late metastasis. A correlation between ultra-late recurrence of melanoma and age at primary diagnosis was not seen. Archived tissue from the primary melanoma was eligible for testing with the immunohistochemical signature (immunoprint^®^) for two patients; it showed a high-risk score and therefore correctly predicted a late recurrence (risk score 0.160, Figure 1). This underlines the potential of predictive signatures used on formalin-fixed paraffin-embedded tissue to filter out these high-risk tumors. It is also consistent with previously reported results from our group showing a very high sensitivity of relapse prediction (>94%) in sentinel-negative patients (stages IB–IIC) with this signature [12]. The immunoprint signature also incorporates CD20 positivity and thereby potentially reflects the pro-metastatic effect of CD20+MSC. Due to the retrospective nature of the study of protracted cases with late or ultra-late recurrent melanomas, the treatment regimens have changed over time. At the time of the primary diagnosis, many patients with higher risk melanomas (≥tumor stage pT1b) might have profited from sentinel lymph node removal and adjuvant therapies. It can be speculated that a fraction of our patients would have been detected early, and late recurrence of melanoma might have been prevented. Moreover, in the future, gene or immunohistochemical signatures can help characterize the tumor microenvironment and identify patients with potential late recurrent melanoma early on. Risk scores from gene or immunohistochemical signatures have the potential to tailor adjuvant therapy independent of the histopathological criteria of the primary melanoma or the sentinel lymph node results. Acceptance rate of adjuvant therapies is high in stage III melanoma patients [37]. Additionally, further follow-up examinations after three years with lymph node sonography and computer tomography should be discussed in patients classified with high-risk scores. Finally, studies with a larger sample size and with control groups are still needed to better understand “tumor dormancy” and the potential co-factors for recurrence in melanoma.

## Figures and Tables

**Figure 1 jcm-11-02026-f001:**
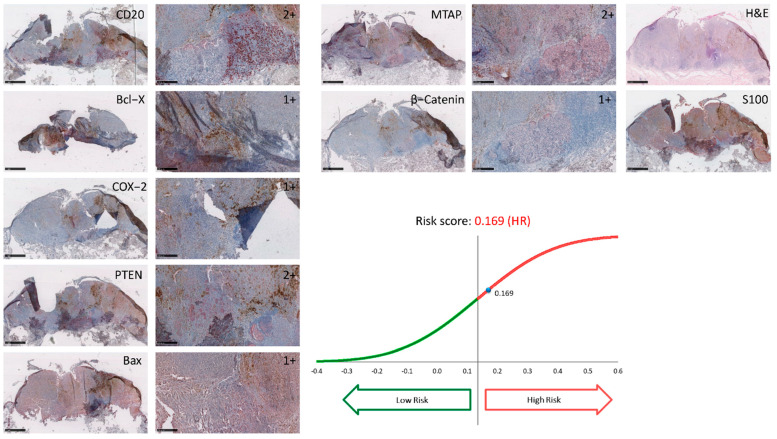
Immunohistochemical staining and risk score stratification Patient A. 1+: weak cytoplasmic staining or less than 20% of cell nuclei stained; 2+: moderate cytoplasmic staining or 21 to 50% of cell nuclei stained. COX-2 = Cyclooxygenase-2, PTEN = Phosphatase and Tensin Homolog, Bax = Bcl-2-associated X protein, MTAP = S-methyl-5′-thioadenosine phosphorylase.

**Figure 2 jcm-11-02026-f002:**
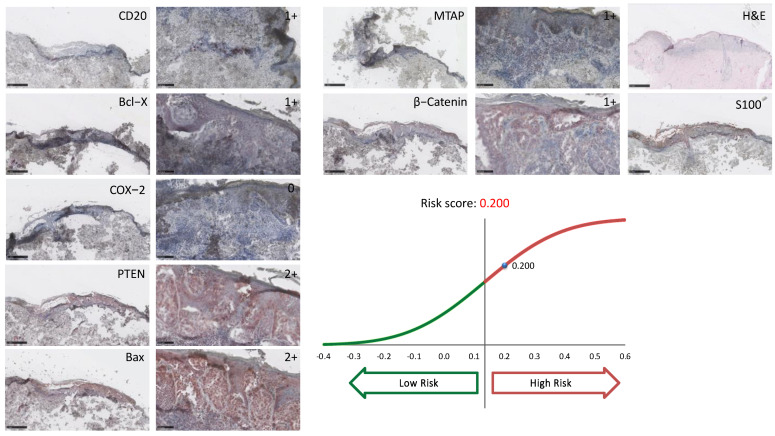
Immunohistochemical staining and risk score stratification Patient B. 0% of cell nuclei stained; 1+: weak cytoplasmic staining or less than 20% of cell nuclei stained; 2+: moderate cytoplasmic staining or 21 to 50% of cell nuclei stained. COX-2 = Cyclooxygenase-2, PTEN = Phosphatase and Tensin Homolog, Bax = Bcl-2-associated X protein, MTAP = S-methyl-5′-thioadenosine phosphorylase.

**Table 1 jcm-11-02026-t001:** Characteristics of study collective, gender comparison.

Overview	Total				Male				Female			
*n* (%)	36 (100%)				18 (50%)				18 (50%)			
	Mean	Median	Maximum	Minimum	Mean	Median	Maximum	Minimum	Mean	Median	Maximum	Minimum
Age at primary diagnosis [years]	51.4	50.6	76	27.8	50.9	51.9	75.8	27.8	51.8	50.6	76	30.2
Age at manifestation of metastatic disease [years]	66.3	68.2	89.4	41	66.7	69.3	89.4	41	65.8	66.4	87.7	41.2
Years until occurrence of metastases	14.9	12.4	33.2	10.1	15.8	13.3	33.2	10.1	14	11.9	29.7	10.1
Localization of primary melanoma:												
Head/Neck *n* (%)	5 (13.9%)				3 (16.7%)				2 (11.1%)			
Trunk *n* (%)	19 (52.8%)				11 (61.1%)				8 (44.4%)			
Upper extremity *n* (%)	5 (13.9%)				1 (5.6%)				4 (22.2%)			
Lower extremity *n* (%)	7 (19.4%)				3 (16.7%)				4 (22.2%)			

**Table 2 jcm-11-02026-t002:** Histopathological parameter of the primary melanoma, gender comparison.

Primary Melanoma Histopathology	Total				Male				Female			
*n* (%)	24 (100%)				12 (50%)				12 (50%)			
Ulceration, *n* (%)	4 (16.7%)				2 (16.7%)				2 (16.7%)			
	Mean	Median	Maximum	Minimum	Mean	Median	Maximum	Minimum	Mean	Median	Maximum	Minimum
Breslow’s depth [mm]	1.8	1.5	4.5	0.4	1.7	1.6	3	0.4	1.8	1.4	4.5	0.6

**Table 3 jcm-11-02026-t003:** Tumor marker S100, gender comparison. Normal S100 vs. increased values. One extraordinary outlier with S100 9.39 ng/mL was excluded from the calculation as it biases the overall picture in this smaller cohort.

S100 at the Time of Metastatic Disease	Total				Male				Female			
*n* (%)	24 (100%)				14 (58.3%)				10 (41.7%)			
S100 normal *n* (%) (<0.105 ng/mL)	14 (58.3%)				10 (71.4%)				4 (40%)			
S100 increased *n* (%) (≥0.105 ng/mL)	10 (41.7%)				4 (28.6%)				6 (60%)			
	Mean	Median	Maximum	Minimum	Mean	Median	Maximum	Minimum	Mean	Median	Maximum	Minimum
S100 (without one outlier) [ng/mL]	0.28	0.07	2.16	0.05	0.25	0.07	2.16	0.05	0.33	0.11	1.85	0.05

**Table 4 jcm-11-02026-t004:** Tumor marker LDH in the entire study collective, gender comparison. Normal LDH vs. increased values. One extraordinary outlier with LDH 21.3µkat/L was excluded from the calculation as it biases the overall picture in this smaller cohort.

LDH at the Time of Metastatic Disease	Total				Male				Female			
*n* (%)	24 (100%)				12 (50%)				12 (50%)			
LDH normal *n* (%) (2.25–3.55 µkat/L)	6 (25%)				4 (33.3%)				2 (16.7%)			
LDH increased *n* (%) (>3.55 µkat/L)	18 (75%)				8 (66.7%)				10 (83.3%)			
	Mean	Median	Maximum	Minimum	Mean	Median	Maximum	Minimum	Mean	Median	Maximum	Minimum
LDH (without one outlier [ng/mL]	5.1	4.7	10.2	2.9	4.8	4.7	8.7	2.9	5.4	4.7	10.2	3.2

**Table 5 jcm-11-02026-t005:** Time period until metastatic disease correlated with neutrophil-lymphocyte-ratio (NLR) at the time of metastasis.

Neutrophil-to-Lymphocyte-Ratio (NLR)		≤5	>5
Time period until metastatic disease [years]	Mean	13.97	19.31
	Median	12.36	15.36

**Table 6 jcm-11-02026-t006:** Comparison of our study collective with cohorts of late recurrence melanoma from 17 publications. * Tsao et al. (1997) investigated only ultra-late recurrence of melanoma (metastatic disease > 15 years after primary diagnosis) ** Percentage refers to the cases with histopathological record (*n* = 24). LR = late recurrence.

	*n* (Total Cases)	% LR Melanoma	*n* (LR Melanoma >10 Years)	*n* (Female)	*n* (Male)	Age at Primary Diagnosis (Years)	Period Until Metastasis (Years)	Breslow’s Depth (mm)	Ulceration	Localization: Trunk	Localization: Lower Extremity	Localization: Upper Extremity	Localization: Head/Neck
Reschke (2022)	-	-	36	18 (50%)	18 (50%)	51	14.9	1.8	4 (16.7%) **	19 (52.8%)	6 (16.7%)	6 (16.7%)	5 (13.9%)
Sarac [20] (2020)	1537		99	45 (45.5%)	54 (54.5%)	-	14.5	-	8 (13.6%)	44 (44.4%)	29 (29.3%)	17 (17.2%)	9 (9.1%)
Osella-Abate [21] (2015)	3580	2.2%	77	46 (59.7%)	31 (40.3%)	48	-	2.3	7 (9.1%)	40 (51.9%)	31 (40.3%)	0	6 (7.8%)
Faries [4] (2013)	-	-	406	171 (41.9%)	235 (57.6%)	41	15.7	1.2	22 (5.4%)	185 (45.3%)	155 (38.0%)		68 (16.7%)
Hohnheiser [22] (2011)	2062	1.6%	34	-	-	52	-	-	-	-	-	-	-
Hansel [23] (2010)	2314	0.9%	20	13 (65%)	7 (35%)	44	13.9	2	4 (20%)	6 (30.0%)	9 (45.0%)	4 (20.0%)	1 (5.0%)
Leman [24] (2003)	-	-	25	13 (52%)	12 (48%)	-	11	-	-	-	-	-	-
Schmid-Wendtner [25] (2000)	6298	0.5%	31	15 (4.4%)	16 (51.6%)	-	12.6	1.4	-	16 (51.6%)	-	-	-
Peters [26] (1997)	-	-	36	20 (55.6%)	16 (44.4%)	50	12.5	1.8	-	12 (33.3%)	11 (30.6%)	5 (13.9%)	8 (22.2%)
Tsao [27] (1997) *	2766	0.7%	20	10 (50%)	10 (50%)	36	18.7	-	-	7 (35.0%)	10 (50.0%)		3 (15.0%)
Tahery [17] (1992)	4301	0.2%	8	3 (37.5%)	5 (62.5%)	43	15.3	-	1 (12.5%)	2 (25.0%)	4 (50.0%)	2 (25.0%)	0
Crowley [28] (1990)	7104	2.4%	168	89 (53.0%)	79 (47.0%)	-	14.3	1.6	-	67 (39.9%)	67 (39.9%)		21 (12.5%)
McEwan [29] (1990)	769	1.4%	11	7 (63.6%)	4 (36.4%)	45	14.2	1.3	2 (18.2%)	3 (27.3%)	3 (27.3%)	3 (27.3%)	2 (18.1%)
Landthaler [30] (1989)	2403	0.4%	10	7 (70%)	3 (30%)	-	-	1.7	-	4 (40.0%)	2 (20.0%)	2 (20.0%)	2 (20.0%)
Gutman [31] (1989)	510	1.1%	6	2 (33.3%)	4 (66.7%)	46	-	-	-	2 (33.3%)	3 (50.0%)	1 (16.7%)	0
Callaway [5] (1989)	-	-	5	5 (100%)	0 (0%)	51	12.9	1.8	0	0	4 (80.0%)		1 (20.0%)
Shaw [32] (1985)	-	-	34	16 (47.1%)	18 (52.9%)	-	13.7	2	-	11 (32.4%)	18 (52.9%)		5 (14.7%)
Briele [33] (1983)	-	-	7	6 (85.7%)	1 (14.3%)	27	16.6	-	-	1 (14.3%)	5 (71.4%)	1 (14.3%)	0
Total			934	441 (49%)	459 (51%)					375 (42.9%)	352 (41.7%)		122 (14.5%)
Average		1.1%				45	14.3	1.7					

## Data Availability

The data presented in this study are available within the article.

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
