# Peer review of "Risk Stratification and Clinical Characteristics of Patients with Late Recurrence of Melanoma (>10 Years)"

_jcm, 2022, doi:10.3390/jcm11072026_

Round 1

Reviewer 1 Report

In the manuscript entitled "Risk stratification and clinical characteristics of patients with late recurrence of melanoma (> 10 years)" the authors suggest that the immunoprint signature may help to identify high-risk primary tumors at the time of initial diagnosis. However, apart from the risk profile of the primary tumor, it seems that the individual immune surveillance can control residual tumor cells for more than a decade. Advanced age and upcoming comorbidities may contribute to the disturbed balance.

The manuscript is well structured, rich in good quality images, the tables are clear. However, there are some points to be revised.

1). The authors report the risk value for the immunoprint signature. We suggest the reference “Reschke R, Gussek P, Ziemer M. Identifying High-Risk Tumors within AJCC Stage IB-III Melanomas Using a Seven-Marker Immunohistochemical Signature. Cancers (Basel). 2021 Jun 10; 13 (12): 2902 ". This paper reports: "the 7-marker signature risk score was calculated as previously described by a linear combination of the marker coefficient and the corresponding IHC measurements normalized by the number of markers measured". The authors could insert the reference to line 88.

2). In lines 173-174 the authors report: the sex of patient B as male / female, and write “d on xxxx. The time between primary diagnosis and metastatic disease was xxx. ”… Is a wrong file attached, not definitive?

Author Response

Dear Reviewer,

thanks a lot for the evaluation and constructive comments. We have answered both points as indicated below and in "track changes" mode in the manuscript.

1). The authors report the risk value for the immunoprint signature. We suggest the reference “Reschke R, Gussek P, Ziemer M. Identifying High-Risk Tumors within AJCC Stage IB-III Melanomas Using a Seven-Marker Immunohistochemical Signature. Cancers (Basel). 2021 Jun 10; 13 (12): 2902 ". This paper reports: "the 7-marker signature risk score was calculated as previously described by a linear combination of the marker coefficient and the corresponding IHC measurements normalized by the number of markers measured". The authors could insert the reference to line 88.

authors' response:

Suggested reference was added in line 88.

2). In lines 173-174 the authors report: the sex of patient B as male / female, and write “d on xxxx. The time between primary diagnosis and metastatic disease was xxx. ”… Is a wrong file attached, not definitive?

authors' response:

Missing information was added in lines 173-174.

Reviewer 2 Report

This is a well-written study titled "Risk stratification and clinical characteristics of patients with late recurrence of melanoma (>10 years)". The authors mostly look into the clinical follow-up on patients with late recurrence of melanoma. It would have been more interesting if there was availability for more tissue for Immunoprint testing to further prove the validity of this test, but it is understandable the lack of available tissue for testing after that duration of time.

The comments I have for the authors are:

  1. In the abstract the duration is stated from 1993 to 2012 while in the materials and methods section it states 1993 to 2015, which duration is correct?
  2. In the abstract should state that only 2 cases had archival tissue for immunoprint testing, and state that in this study it is difficult to draw a conclusion given the minimal number of cases tested.
  3. In the introduction line 35 Clark's level should be removed from a predictor of prognosis as it has been proven to not be beneficial and poorly reproducible between dermatopathologists.
  4. page 5 line 172: there are missing data on patient B (Gender, location, time between primary and metastatic disease).

Author Response

Dear Reviewer,

thanks a lot for the evaluation and constructive critique. We have answered every single point. See in manuscript indicated with "track changes" and answers below.

In the abstract the duration is stated from 1993 to 2012 while in the materials and methods section it states 1993 to 2015, which duration is correct?

Author's response: You are absolutely correct. That was a typo and changed into 2012 in methods. See line 77.

In the abstract should state that only 2 cases had archival tissue for immunoprint testing, and state that in this study it is difficult to draw a conclusion given the minimal number of cases tested.

Authors' response: We added that information to line 19,20.

In the introduction line 35 Clark's level should be removed from a predictor of prognosis as it has been proven to not be beneficial and poorly reproducible between dermatopathologists.

Authors' response: Clark's level was removed in line 35.

page 5 line 172: there are missing data on patient B (Gender, location, time between primary and metastatic disease).

Author's response: missing data was added in lines 180-182.

We hope that the changes made due to the reviewers suggestions renders the article publishable. 

Kindest regards

Dr. Reschke